# Evolution of Molar Mass Distributions Using a Method of Partial Moments: Initiation of RAFT Polymerization

**DOI:** 10.3390/polym14225013

**Published:** 2022-11-18

**Authors:** Charles H. J. Johnson, Thomas H. Spurling, Graeme Moad

**Affiliations:** 1CSIRO Manufacturing, Clayton, VIC 3168, Australia; 2Centre for Transformative Innovation, School of Business, Law and Entrepreneurship, Swinburne University of Technology, Hawthorn, VIC 3122, Australia

**Keywords:** RAFT polymerization, reversible addition-fragmentation chain transfer, SUMI (single-unit monomer insertion), reversible deactivation radical polymerization (RDRP), kinetic simulation, molar mass distribution

## Abstract

We describe a method of partial moments devised for accurate simulation of the time/conversion evolution of polymer composition and molar mass. Expressions were derived that enable rigorous evaluation of the complete molar mass and composition distribution for shorter chain lengths (e.g., degree of polymerization, *X*_n_ = *N* < 200 units) while longer chains (*X*_n_ ≥ 200 units) are not neglected, rather they are explicitly considered in terms of partial moments of the molar mass distribution, $\mu_{x}^{N}(P)=\sum_{n=N+1}^{\infty }n^{x}[P_{n}]$ (where P is a polymeric species and *n* is its’ chain length). The methodology provides the exact molar mass distribution for chains *X*_n_ < *N,* allows accurate calculation of the overall molar mass averages, the molar mass dispersity and standard deviations of the distributions, provides closure to what would otherwise be an infinite series of differential equations, and reduces the stiffness of the system. The method also allows for the inclusion of the chain length dependence of the rate coefficients associated with the various reaction steps (in particular, termination and propagation) and the various side reactions that may complicate initiation or initialization. The method is particularly suited for the detailed analysis of the low molar mass portion of molar mass distributions of polymers formed by radical polymerization with reversible addition-fragmentation chain transfer (RAFT) and is relevant to designing the RAFT-synthesis of sequence-defined polymers. In this paper, we successfully apply the method to compare the behavior of thermally initiated (with an added dialkyldiazene initiator) and photo-initiated (with a RAFT agent as a direct photo-iniferter) RAFT-single-unit monomer insertion (RAFT-SUMI) and oligomerization of *N*,*N*-dimethylacrylamide (DMAm).

## 1. Introduction

The last 25 years have seen the emergence of reversible deactivation radical polymerizations (RDRP) [1,2,3]. These processes possess many of the attributes of classical living polymerization (i.e., ability to chain extend, molar mass control, low molar mass dispersity, and the ability to synthesize blocks and complex architectures) [4,5] and yet have much of the versatility (i.e., broad monomer scope, compatibility with a wide range of reaction conditions) associated with conventional radical polymerization. For RDRP to be successful, and for living characteristics exhibited, a mechanism for rapidly and reversibly activating and deactivating the propagating species is required. This mechanism provides a means of chain equilibration and allows an acceptable rate of polymerization to be achieved while maintaining the concentration of reactive intermediates at a sufficiently low level for kinetic stability. 

IUPAC distinguish three basic types of RDRP according to the mechanism of the activation-deactivation process. These are (1) RDRP with unimolecular activation by dissociation of an initiator, known as stable radical-mediated polymerization (SRMP) [2,6], (2) RDRP with bimolecular activation by reaction of an initiator with an activator, known as atom-transfer radical polymerization (ATRP), and (3) RDRP with activation by degenerative chain transfer, known as degenerative-chain-transfer radical polymerization (DTRP). In each case deactivation is simply the reverse of the activation process. Reversible addition-fragmentation chain transfer (RAFT) polymerization is a subclass of DTRP where the reversible transfer step involves the formation of a transient intermediate. In some polymerizations, more than one of the three basic mechanisms (SRMP, ATRP and DTRP) may operate simultaneously. For example, in some RAFT polymerizations, the radicals required to maintain the process are formed directly from the RAFT agent by unimolecular dissociation, for example, by photolysis [7,8,9], by bimolecular dissociation in a redox or electrochemical (eRAFT) process [10,11,12], or by a photosensitized or photoredox reaction such as photo-induced electron or energy transfer-RAFT (PET-RAFT) [8]. Many examples can be found in recent reviews [3,13,14,15,16].

Many have sought to model the kinetics of RAFT and other RDRP with a view to predicting the conversion-time profile for polymer products, molar mass distributions, end-group fidelity, and/or copolymer compositions [17,18]. A major preoccupation, in modelling RAFT polymerization, has been to examine the effect of different mechanisms on polymerization kinetics and molar mass distributions with the aim of understanding the retardation that is often manifest [19,20,21]. Various approaches, both deterministic (kinetic simulation) and stochastic methods (Monte Carlo simulation), have been applied, most often with some significant simplification of the mechanism. 

Moad et al. [22] conducted the first kinetic simulation of RAFT polymerization to predict molar mass distributions when they applied a method of partial moments to model RAFT polymerization of methyl methacrylate (MMA) mediated by macromonomer RAFT agents [23,24] (the process more recently called sulfur-free RAFT [25,26]). The method of partial moments was subsequently also successfully applied to dithiobenzoate-mediated RAFT polymerization of styrene and MMA [27]. The approach used involved successfully solving the complete set of differential equations for all species up to some “cut-off” value (*N*) of the chain length (e.g., *N* ≤ 100 units) with higher molar mass species (*N* > 100 units) not being neglected but defined as their partial moments ($\mu_{x}^{N}(P)$, Figure 1, Equations (1) and (2)). The evolution of the molar mass distribution as a function of time/conversion was accurately modelled within these constraints.
(1)$$\mu_{x}(P)=\sum_{n=1}^{\infty }n^{x}[P_{n}]$$
$$\;=\sum_{n=1}^{N}n^{x}[P_{n}]+\sum_{n=N+1}^{\infty }n^{x}[P_{n}]$$
$$\;=\sum_{n=1}^{N}n^{x}[P_{n}]+\mu_{x}^{N}(P)$$
(2)$$\mathrm{where}\;\mu_{x}^{N}(P)=\sum_{n=N+1}^{\infty }n^{x}[P_{n}]$$

We had previously applied this same methodology to aminoxyl [nitroxide]-mediated polymerization (NMP) [28] – a form of SRMP. That study [15] provided the first (virtual) demonstration of the potential of NMP to produce low dispersity polymers [29]. We then applied our method in the kinetic simulation of RAFT polymerization of MMA mediated by 2-cyanopropan-2-yl benzodithioate, We reported on the results of kinetic simulation of this system using a similar methodology in 2003 [27] as a means of estimating the transfer coefficients of the RAFT agents, but provided no details of the simulation method at that time.

Methods for mathematical modelling RAFT polymerization have been reviewed [17,30,31]. In RDRP, and in particular. RAFT polymerization, the calculation of full molar mass distributions is made more complex by a larger number of different reactive polymeric species. In addition to the propagating species (P*_n_*•), and the dead chains formed in termination by combination (P^C^*_m_*_+*n*_) or disproportionation (P^H^*_m_* and P^=^*_n_*), there are the macroRAFT agent (P*_n_*Z), and the various intermediates (P*_n_*ŻR, P*_n_*ŻI and P*_n_*ŻP*_m_*) (refer to Figure 2). The system of equations is substantially further expanded when we include intermediate-radical termination and the products from that process in the simulation.

In order to limit the size of the simulation, Moad et al. [22,27] treated the intermediates P*_n_*ŻP*_m_* as two independent molar mass distributions, such that the main equilibrium (reactions 20 and 21) is effectively represented by (Figure 3, Reactions 40–42). This same strategy was later implemented in kinetic simulation of RAFT polymerization when using the commercial Predici^®^ package [32,33,34,35,36,37,38,39,40,41]. The products arising from intermediate radical termination can in principle be similarly modelled by considering three or four independent distributions as appropriate [42]. It is also possible to reconstruct the full molar mass distribution of the species that comprise multiple distributions by Monte Carlo simulation [43]. Full molar mass distributions have been obtained without this assumption when making use of a quasi-steady state approximation [44,45], which has the effect of removing the active species from direct consideration and thereby the stiffness of the differential equations is reduced. 

Other strategies developed for modelling molar mass distributions produced by RAFT polymerization include use of the Monte Carlo simulation [46,47,48,49], coarse graining [50], and the use of probability generating functions [51]. In implementing these simulation methods, various simplifications are commonly introduced. These include:Replacing the main addition-fragmentation chain equilibria by a simpler degenerative substitution chain transfer process (no intermediate involved, i.e., Figure 4 [52].Making the pre-“equilibrium” irreversible (or neglecting the pre-equilibrium) [53]Ignoring intermediate radical termination (many papers).Ignoring (irreversible) termination [54].The use of a quasi-steady state approximation [44,45,55].

These simplifications can all be justified in special circumstances, but they do not have general applicability. They are a particular concern when modelling the detailed molar mass distribution of low molar mass polymers, or the low molar mass portion of high polymers, and are not appropriate in the case of the examples described later in this paper. 

Mention should also be made of the method of moments, which has been used to model RAFT polymerization [36,56,57], but does not directly give molar mass distributions. A very comprehensive study of RAFT polymerization of methyl acrylate with cumyl and 2-cyanopropan-2-yl dithiobenzoate has recently been reported by Zapata-Gonzalez et al [57]. The method of moments is described in Appendix A.

Additional reactions considered with respect to assessing the importance of intermediate radical termination are shown in Figure 5. Of these, Reaction (58) was found to be of significance when the rate coefficient exceeded 10^8^ M^−1^ s^−1^. The other reactions were found to be of little significance even with a rate coefficient of 10^9^ M^−1^ s^−1^. For the polymerizations considered here, fragmentation is fast, and the concentration of intermediate radicals is correspondingly low. This is consistent with intermediate radical termination being essentially unknown in trithiocarbonate-mediated polymerization of most more-activated monomers (MAMs, which include (meth)acrylates, (meth)acrylamides, styrenes) or in polymerization of MMA mediated by dithiobenzoates [58]. Intermediate radical termination becomes of greater significance when rates of intermediate fragmentation are low or values of the reverse transfer constant (*C*_-tr_ = *k_-β_/k*_iR_) are high and can be of greater importance in single unit monomer insertion (SUMI) or oligomerization experiments. The intermediate radical termination processes are shown as occurring by combination but might also involve disproportionation. For the case of dithiobenzoate-mediated RAFT polymerization an additional set of reactions corresponding to the so-called “missing step” processes should also be considered [57,59].

## 2. Materials and Methods 

The experimental procedures and materials and instrumentation used are described in our previous papers [27,42,60,61]. The synthesis of 4-cyano-4-(((ethylthio)carbonothioyl)thio)pentanoic acid (**1**) is also described elsewhere [62].

### 2.1. Thermally Initiated RAFT Oligomerization of N,N-dimethylacrylamide (DMAm)

DMAm, (0.099 g, 1.000 mmol), the trithiocarbonate **1** (0.132 g, 0.50 mmol) and Na_2_CO_3_ (0.024 g, 0.226 mmol) were placed in small 5 mL vial and dissolved in 0.5 mL D_2_O. the initiator, VA-044 (0.025 g, 0.078 mmol) was then added and the resultant solution was transferred to a flame sealable NMR tube. The solution was degassed by 3 freeze-pump-thaw cycles and NMT tube sealed under vacuum. The exact amounts of monomer, trithiocarbonate and initiator present were determined by an initial ambient temperature NMR spectrum. The oligomerization was initiated by inserting the NMR tube into probe of the NMR that had been preheated to 60 °C. After 4.5 h at 60 °C the reaction was quenched by removing the NMR tube from the spectrometer and rapid cooling. The average degree of polymerization, calculated from the final ^1^H-NMR spectra, was 2.19. The evolution of species vs time observed by in situ NMR is shown in Figure 1a.

Similar reactions prepared with degassing by sparging with ultra-pure nitrogen gave a short variable (up to 30 min) inhibition period. No discernable inhibition period was observed in the present experiments.

### 2.2. Photochemically Initiated RAFT Oligomerization of N,N-dimethylacrylamide (DMAm)

Conditions for RAFT oligomerization of DMAm in presence of trithiocarbonate **1** in 0.045 M Na_2_CO_3_ in D_2_O with direct photoinitiation are reported elsewhere [61]. Polymerization mixtures were prepared as for thermally initiated experiments (Section 2.1), but no initiator was added. Initiating radicals were generated directly from the RAFT agent by irradiation with a blue (451 nm) light from a light-emitting diode (LED) source. The reaction temperature was maintained at 65 °C. The sample was removed from the photoreactor and rapidly cooled for NMR measurements at ambient temperature. Small amounts of by-products, <2% overall yield, attributable to self-reaction of **7** (R∙) were observed [63]. The evolution of species vs time observed by in situ NMR is shown in Figure 2a.

### 2.3. Kinetic Simulation 

The differential equations were solved numerically using the function NDSolve function in Mathematica (Version 13.1, Wolfram Research, Inc., Champaign, Illinois, 2022) with default parameter settings. The calculations were performed with a Dell Latitude E7470 laptop with an Intel^®^ Core^TM^ i7-6600U CPU @ 2.60 GHz, 8 GB RAM and Windows 10 (64 bit) operating system. A typical value for Absolute Timing for solving the differential equations with a chain length (*N*) of 5 monomer units was 0.45 s, with *N* = 50 was 2.41 s, with *N* = 100 was 9.46 s, and with *N* = 200 was 129.06 s. The solution with *N ≥ *200 required the additional NDSolve option: Method -> {"EquationSimplification” -> "Residual"}.

## 3. Results and Discussion

Some of the structures and the corresponding structure numbers and symbols used are provided in Figure 3.

### 3.1. Differential Equations

The reactions included in the present simulations are listed in Figure 2. The differential equations for low molar mass species are simply derived. Those for the initiator-derived (I•) and initial RAFT agent-derived radicals (R•) are shown below (Section 3.1.1 and Section 3.1.2, respectively). When a reaction involved a polymeric species (P), terms relating to the partial moments $\mu_{x}^{N}(P)$ were introduced. The differential equations for the reactions of polymeric species where there is a change in molar mass (specifically propagation, irreversible termination by combination, reversible addition-fragmentation chain transfer) are more complex when they involve partial moments. Details of these expressions are provided in Section 3.1.3, Section 3.1.4 and Section 3.1.5, respectively.

Direct photoinitiation by photolysis of the RAFT agent is covered by the inclusion of Reactions (31–41) shown in Figure 2. It was found necessary to include the self-reaction of the thiocarbonylthio radical (Z•) forming the disulfide as a photochemically reversible process (*vide infra*).

#### 3.1.1. Initiator-Derived Radicals (I∙)

The differential equation associated with initiator-derived radicals (I∙) formed from an exogenous initiator (I_2_) is shown in Equation (3). Note that Reaction (4, Figure 2) covers loss of initiator by the cage reaction of the initiator-derived radicals I∙, *f_g_* is the efficiency for radical generation [64]. The encounter reactions of I∙, which are important in determining the efficiency for initiation of polymerization (*f*_i_), are embraced in Reaction (2) (for further detail on initiation mechanisms and initiator efficiencies see [64]).
(3)$$\begin{array}{ll} \frac{d[I\cdot ]}{dt} & =\overset{\mathrm{initiation}}{2k_{d}f_{g}[I_{2}]-k_{i}[I\cdot ][M]}+\overset{\mathrm{photoinitiation}}{\left(k_{\mathrm{dzi}}^{}[\mathrm{IZ}]-k_{\mathrm{tzi}}^{}[I\cdot ][Z\cdot ]\right)}-\overset{\mathrm{transfer}\;\mathrm{to}\;\mathrm{disulfide}}{k_{\mathrm{triz}2}^{}[I\cdot {][Z}_{2}]}\\ & +\overset{\mathrm{reaction}\;\mathrm{of}\;I\cdot \;\mathrm{with}\;\mathrm{initial}\;\mathrm{RAFT}\;\mathrm{agent}}{\left(-k_{a,\mathrm{IR}}[I\cdot ][\mathrm{RZ}]+k_{\beta ,\mathrm{IR}}[R\overset{\cdot }{Z}I]\right)}\;\;\;\;+\overset{\mathrm{reaction}\;\mathrm{of}\;I\cdot \;\mathrm{with}\;\mathrm{initiator}-\mathrm{derived}\;\mathrm{RAFT}\;\mathrm{agent}}{\left(-k_{a,\mathrm{IR}}[I\cdot ][\mathrm{IZ}]+2k_{\beta ,\mathrm{IR}}[I\overset{\cdot }{Z}I]\right)}\\ & +\sum_{n=1}^{\infty }\overset{\mathrm{reaction}\;\mathrm{of}\;I\cdot \;\mathrm{with}\;\mathrm{macroRAFT}\;\mathrm{agent}}{\left(-k_{-\beta ,I}^{}[P_{n}Z][I\cdot ]+k_{\beta ,I}^{}[P_{n}\overset{\cdot }{Z}I]\right)}\\ & -\overset{\mathrm{reactions}\;\mathrm{between}\;\mathrm{small}\;\mathrm{radicals}}{2k_{\mathrm{prt}}^{1}{\left[I\cdot \right]}^{2}-k_{\mathrm{prt}}^{1}[I\cdot ][R\cdot ]}\;\;\;\;-\overset{\mathrm{primary}\;\mathrm{radical}\;\mathrm{termination}}{[I\cdot ]\sum_{n=1}^{\infty }k_{prt}^{n}[P_{n}\cdot ]} \end{array}$$

When partial moments are used, the terms involving polymeric species, shown in blue or red in Equation (3), are replaced as indicated in Equations (4) or (5), respectively.
(4)$$\begin{array}{l} \sum_{n=1}^{\infty }\left(-k_{-\beta ,I}^{}[P_{n}Z][I\cdot ]+k_{\beta ,I}^{}[P_{n}\overset{\cdot }{Z}I]\right)=\\ \;\;\;\;\;\;\;\;-k_{-\beta ,I}^{}[I\cdot ]\left(\sum_{n=1}^{N}\left[P_{n}Z]+\mu_{0}^{N}(P_{n}Z\right)\right)+k_{\beta ,I}^{}\left(\sum_{n=1}^{N}\left[P_{n}\overset{\cdot }{Z}I\right]+\mu_{0}^{N}(P_{n}\overset{\cdot }{Z}I)\right) \end{array}$$
(5)$$[I\cdot ]\sum_{n=1}^{\infty }k_{prt}^{n}[P_{n}\cdot ]=[I\cdot ]\left(\sum_{n=1}^{N}k_{prt}^{n}[P_{n}\cdot ]+k_{prt}^{N}\mu_{0}^{N}(P_{n}\cdot )\right)$$

#### 3.1.2. Initial RAFT Agent-Derived Radicals (R∙)

The expression describing radicals (R∙) derived from the initial RAFT agent is similar and is shown in Equation (6).
(6)$$\begin{array}{ll} \frac{d[R\cdot ]}{dt} & =-\overset{\mathrm{initiation}}{k_{i,R}[R\cdot ][M]}+\overset{\mathrm{reversible}\;\mathrm{photoinitiation}}{\left(k_{\mathrm{dzr}}^{}[\mathrm{RZ}]-k_{\mathrm{tzr}}^{}[R\cdot ][Z\cdot ]\right)}-\overset{\mathrm{transfer}\;\mathrm{to}\;\mathrm{disulfide}}{k_{\mathrm{trrz}2}^{}[R\cdot {][Z}_{2}]}\\ & \;\;+\overset{\mathrm{reaction}\;\mathrm{with}\;\mathrm{initial}\;\mathrm{RAFT}\;\mathrm{agent}}{\left(-k_{a,\mathrm{RR}}[R\cdot ][\mathrm{RZ}]+2k_{\beta ,\mathrm{RR}}[R\overset{\cdot }{Z}R]\right)}\;\;+\;\;\;\left(\overset{\mathrm{reaction}\;\mathrm{with}\;\mathrm{initiator}-\mathrm{derived}\;\mathrm{RAFT}\;\mathrm{agent}}{-k_{a,\mathrm{IR}}[R\cdot ][\mathrm{IZ}]+k_{\beta ,\mathrm{IR}}[R\overset{\cdot }{Z}I]}\right)\\ & \;\;+\sum_{n=1}^{\infty }\overset{\mathrm{reaction}\;\mathrm{with}\;\mathrm{macroRAFT}\;\mathrm{agent}}{\left(-k_{-\beta ,R}^{}[P_{n}Z][R\cdot ]+k_{\beta ,R}^{}[P_{n}\;\overset{\cdot }{Z}R]\right)}\\ & \;\;-\overset{\mathrm{reaction}\;\mathrm{between}\;\mathrm{small}\;\mathrm{radicals}}{2k_{\mathrm{prt}}^{1}{\left[R\cdot \right]}^{2}-k_{\mathrm{prt}}^{1}[I\cdot ][R\cdot ]}\;\;\;\;-\overset{\mathrm{primary}\;\mathrm{radical}\;\mathrm{termination}}{[R\cdot ]\sum_{n=1}^{\infty }k_{prt}^{n}[P_{n}\cdot ]} \end{array}$$

Again, when partial moments are used, the terms involving polymeric species (shown in blue or red) are replaced by the expressions shown in Equations (7) and (8), respectively.
(7)$$\begin{array}{l} \sum_{n=1}^{\infty }\left(-k_{-\beta ,I}^{}[P_{n}Z][R\cdot ]+k_{\beta ,I}^{}[P_{n}\overset{\cdot }{Z}R]\right)=\\ \;\;-k_{-\beta ,I}^{}[R\cdot ]\left(\sum_{n=1}^{N}\left[P_{n}Z]+\mu_{0}^{N}(P_{n}Z\right)\right)+k_{\beta ,I}^{}\left(\sum_{n=1}^{N}\left[P_{n}\overset{\cdot }{Z}R\right]+\mu_{0}^{N}(P_{n}\overset{\cdot }{Z}R)\right) \end{array}$$
(8)$$[R\cdot ]\sum_{n=1}^{\infty }k_{prt}^{n}[P_{n}\cdot ]=[R\cdot ]\left(\sum_{n=1}^{N}k_{prt}^{n}[P_{n}\cdot ]+k_{prt}^{N}\mu_{0}^{N}(P_{n}\cdot )\right)$$

#### 3.1.3. Propagating Radicals (P_n_∙)

The differential equation for the unimer propagating species (P_1_∙) is shown in Equation (9).
(9)$$\begin{array}{ll} \frac{d[P_{1}\cdot ]}{dt} & =\overset{\mathrm{initiation}}{k_{i}[I\cdot ][M]+k_{i,R}[R\cdot ][M]}-\overset{\mathrm{propagation}}{k_{p}[P_{1}\cdot ][M]}\\ & +\overset{\mathrm{reversible}\;\mathrm{photoinitiation}}{\left(k_{\mathrm{dzp}}^{}[P_{1}Z]-k_{\mathrm{tzp}}^{}[P_{1}\cdot ][Z\cdot ]\right)}\;-\;\;\overset{\mathrm{transfer}\;\mathrm{to}\;\mathrm{disulfide}}{k_{\mathrm{trpZ}2}^{}[P_{1}\cdot {][Z}_{2}]}\\ & +\overset{\mathrm{reaction}\;\mathrm{with}\;\mathrm{initial}\;\mathrm{RAFT}\;\mathrm{agent}}{\left(-k_{a,R}^{}[\mathrm{RZ}][P_{1}\cdot ]+k_{-a,R}^{}[P_{1}\overset{\cdot }{Z}R]\right)}\;\;+\;\;\;\overset{\mathrm{reaction}\;\mathrm{with}\;\mathrm{initiator}-\mathrm{derived}\;\mathrm{RAFT}\;\mathrm{agent}}{\left(-k_{a,R}^{}[\mathrm{IZ}][P_{1}\cdot ]+k_{-a,R}^{}[P_{1}\overset{\cdot }{Z}I]\right)}\\ & -\overset{\mathrm{reaction}\;\mathrm{with}\;\mathrm{macroRAFT}\;\mathrm{agent}}{k_{a}^{}[P_{1}\cdot ]\left(\sum_{m=1}^{N}\left[P_{m}Z\right]+\mu_{0}^{}(\mathrm{PZ})\right)}\;\;+\overset{\mathrm{intermediate}\;\mathrm{fragmentation}}{\;\;0.5k_{\beta }^{}([P_{1}\;\overset{\cdot }{Z}P]+[{P\overset{\cdot }{Z}P}_{1}])}\\ & -\overset{\mathrm{termination}}{[P_{1}\cdot ]\left(\sum_{m=1}^{N}k_{\mathrm{tc}+\mathrm{td}}^{1,m}[P_{n}\cdot ]+k_{\mathrm{tc}+\mathrm{td}}^{1,N}\mu_{0}^{N}(P\cdot )\right)} \end{array}$$

The differential equations for propagating species of lengths 2 < *n* ≤ *N* are similar (Equation (10)).
(10)$$\begin{array}{ll} \frac{d[P_{n}\cdot ]}{dt} & =\overset{\mathrm{propagation}}{k_{p}[P_{n-1}\cdot ][M]-k_{p}[P_{n}\cdot ][M]}+\overset{\mathrm{reversible}\;\mathrm{photoinitiation}}{\left(k_{\mathrm{dzp}}^{}[P_{n}Z]-k_{\mathrm{tzp}}^{}[P_{n}\cdot ][Z\cdot ]\right)}-\overset{\mathrm{transfer}\;\mathrm{to}\;\mathrm{disulfide}}{k_{\mathrm{trpZ}2}^{}[P_{n}\cdot {][Z}_{2}]}\\ & \overset{\mathrm{reaction}\;\mathrm{with}\;\mathrm{initial}\;\mathrm{RAFT}\;\mathrm{agent}}{+\left(-k_{a,R}^{}[\mathrm{RZ}][P_{n}\cdot ]+k_{-a,R}^{}[P_{n}\overset{\cdot }{Z}R]\right)}\;\;+\overset{\mathrm{reaction}\;\mathrm{with}\;\mathrm{initiator}-\mathrm{derived}\;\mathrm{RAFT}\;\mathrm{agent}}{\;\;\left(-k_{a,R}^{}[\mathrm{IZ}][P_{n}\cdot ]+k_{-a,R}^{}[P_{n}\overset{\cdot }{Z}I]\right)}\\ & -\overset{\mathrm{reaction}\;\mathrm{with}\;\mathrm{macroRAFT}\;\mathrm{agent}}{k_{a}^{}[P_{n}\cdot ]\left(\sum_{m=1}^{N}\left[P_{m}Z\right]+\mu_{0}^{}(\mathrm{PZ})\right)}\;\;+\;\;\;\overset{\mathrm{intermediate}\;\mathrm{fragmentation}}{0.5k_{\beta }^{}([P_{n}\;\overset{\cdot }{Z}P]+[{P\overset{\cdot }{Z}P}_{n}])}\\ & -\overset{\mathrm{termination}}{[P_{n}\cdot ]\left(\sum_{m=1}^{N}k_{\mathrm{tc}+\mathrm{td}}^{n,m}[P_{n}\cdot ]+k_{\mathrm{tc}+\mathrm{td}}^{n,N}\mu_{0}^{N}(P\cdot )\right)}\\ & -\overset{\mathrm{primary}\;\mathrm{radical}\;\mathrm{termination}}{k_{\mathrm{prt}}^{n}[P_{n}\cdot ][I\cdot ]-k_{\mathrm{prt}}^{n}[P_{n}\cdot ][R\cdot ]} \end{array}$$

In the case of the differential equation for the partial moments of the propagating species describing lengths *n* > *N*, $\mu_{x}^{N}(P\cdot )$, the term relating to the propagation reaction (Reaction 4, Figure 2) can be expanded and simplified as shown in Equation (11).
(11)$$\begin{array}{l} -k_{p}^{N}[M]\left(\sum_{n=N+1}^{\infty }n^{x}[P_{n}\cdot ]-\sum_{n=N+1}^{\infty }n^{x}[P_{n-1}\cdot ]\right)=\\ \;\;\;\;\;\;\;\;\;\;-k_{p}^{N}[M]\left(\mu_{x}^{N}(P\cdot )-{\left(N+1\right)}^{x}[P_{N}\cdot ]-\sum_{r=0}^{x}\left(\begin{array}{c} x\\ r \end{array}\right)\mu_{r}^{N}(P\cdot )\right) \end{array}$$

The complete expression for the differential equation for the partial moments $\mu_{x}^{N}(P\cdot )$ is then as shown in Equation (12).
(12)$$\begin{array}{ll} \frac{d\mu_{x}^{N}(P\cdot )}{dt} & =-\overset{\mathrm{propagation}}{k_{p}^{N}[M]\left(\mu_{x}^{N}(P\cdot )-{\left(N+1\right)}^{x}[P_{N}\cdot ]-\sum_{r=0}^{x}\left(\begin{array}{c} x\\ r \end{array}\right)\mu_{r}^{N}(P\cdot )\right)}\\ & +\overset{\mathrm{reversible}\;\mathrm{photoinitiation}}{\left(k_{\mathrm{dzp}}^{}\mu_{x}^{N}(\mathrm{PZ})-k_{\mathrm{tzp}}^{}\mu_{x}^{N}(P\cdot )[Z\cdot ]\right)}-\overset{\mathrm{transfer}\;\mathrm{to}\;\mathrm{disulfide}}{k_{\mathrm{trpZ}2}^{}\mu_{x}^{N}(P\cdot ){[Z}_{2}]}\\ & -\overset{\mathrm{Reaction}\;\mathrm{with}\;\mathrm{initial}\;\mathrm{RAFT}\;\mathrm{agent}}{k_{a,R}^{N}[\mathrm{RZ}]\mu_{x}^{N}(P\cdot )+k_{-a,R}^{N}\mu_{x}^{N}(P\overset{\cdot }{Z}R)}\\ & -\overset{\mathrm{Reaction}\;\mathrm{with}\;\mathrm{initiator}-\mathrm{derived}\;\mathrm{RAFT}\;\mathrm{agent}}{k_{a,I}^{N}[\mathrm{IZ}]\mu_{x}^{N}(P\cdot )+k_{-a,I}^{N}\mu_{x}^{N}(P\overset{\cdot }{Z}I)}\\ & +\overset{\mathrm{intermediate}\;\mathrm{fragmentation}}{0.5k_{\beta }^{}\left(\mu_{x}^{N}(P_{n}\overset{\cdot }{Z}P)+\mu_{x}^{N}({P\;\overset{\cdot }{Z}P}_{n})\right)}\;\;+\;\;\;0.5k_{\beta }^{}\left(\mu_{x}^{N}(P_{n}\;\overset{\cdot }{Z}P)+\mu_{x}^{N}({P\overset{\cdot }{Z}P}_{n})\right)\\ & -\overset{\mathrm{termination}}{\mu_{x}^{N}(P\cdot )\left(\sum_{n=1}^{N}k_{tc+td}^{n,N}[P_{n}\cdot ]+k_{tc+td}^{N,N}\mu_{0}^{N}(P\cdot )\right)}\\ & -\overset{\mathrm{primary}\;\mathrm{radical}\;\mathrm{termination}}{k_{\mathrm{prt}}^{N}[I\cdot ]\mu_{x}^{N}(P\cdot )-k_{\mathrm{prt}}^{N}[R\cdot ]\mu_{x}^{N}(P\cdot )} \end{array}$$

#### 3.1.4. Initial RAFT Agent (RZ)

The expression describing reactions of the initial RAFT agent (RZ) is shown in Equation (13).
(13)$$\begin{array}{ll} \frac{d[RZ]}{dt} & =\overset{\mathrm{reversible}\;\mathrm{photoinitiation}}{\left(k_{\mathrm{tzr}}^{}[R\cdot ][Z\cdot ]-k_{\mathrm{dzr}}^{}[\mathrm{RZ}]\right)}+\overset{\mathrm{transfer}\;\mathrm{to}\;\mathrm{disulfide}}{k_{\mathrm{trrz}2}^{}[R\cdot {][Z}_{2}]}\\ & \overset{\mathrm{Reaction}\;\mathrm{of}\;R\cdot \;\mathrm{with}\;\mathrm{initial}\;\mathrm{RAFT}\;\mathrm{agent}}{+\left(-k_{a,\mathrm{RR}}[R\cdot ][\mathrm{RZ}]+2k_{\beta ,\mathrm{RR}}[R\overset{\cdot }{Z}R]\right)}\;+\overset{\mathrm{Reaction}\;\mathrm{of}\;I\cdot \;\mathrm{with}\;\mathrm{initial}\;\mathrm{RAFT}\;\mathrm{agent}}{\left(-k_{a,\mathrm{IR}}[I\cdot ][\mathrm{RZ}]+k_{\beta ,\mathrm{IR}}[I\overset{\cdot }{Z}R]\right)}\\ & +\overset{\mathrm{reaction}\;\mathrm{of}\;P_{1}\cdot \;\mathrm{with}\;\mathrm{initial}\;\mathrm{RAFT}\;\mathrm{agent}}{\left(-k_{a,R}^{}[\mathrm{RZ}][P_{1}\cdot ]+k_{-a,R}^{}[P_{1}\overset{\cdot }{Z}R]\right)}\\ & +\overset{\mathrm{reaction}\;\mathrm{of}\;P_{n}\cdot \;\mathrm{with}\;\mathrm{initial}\;\mathrm{RAFT}\;\mathrm{agent}}{\left(-k_{a,R}^{}[\mathrm{RZ}][P_{n}\cdot ]+k_{-a,R}^{}[P_{n}\overset{\cdot }{Z}R\right)} \end{array}$$

#### 3.1.5. MacroRAFT Agent (P_n_Z)

The differential equations for macroRAFT agents of lengths 1 < n ≤ N are shown in Equation (14).
(14)$$\begin{array}{ll} \frac{d[P_{n}Z]}{dt} & =\overset{\mathrm{reversible}\;\mathrm{photoinitiation}}{\left(-k_{\mathrm{dzp}}^{}[P_{n}Z]+k_{\mathrm{tzp}}^{}[P_{n}\cdot ][Z\cdot ]\right)}+\overset{\mathrm{transfer}\;\mathrm{to}\;\mathrm{disulfide}}{k_{\mathrm{trpZ}2}^{}[P_{n}\cdot {][Z}_{2}]}\\ & \overset{\mathrm{reaction}\;\mathrm{of}\;I\cdot \;\mathrm{with}\;\mathrm{macroRAFT}\;\mathrm{agent}}{-k_{-\beta ,I}^{}[P_{n}Z][I\cdot ]+k_{\beta ,I}^{}[P_{n}\overset{\cdot }{Z}I]}-\overset{\;\;\;\;\mathrm{reaction}\;\mathrm{of}\;R\cdot \;\mathrm{with}\;\mathrm{macroRAFT}\;\mathrm{agent}}{k_{-\beta ,R}^{}[P_{n}Z][R\cdot ]+k_{\beta ,R}^{}[P_{n}\overset{\cdot }{Z}R]}\\ & -\overset{\mathrm{reaction}\;\mathrm{with}\;\mathrm{macroRAFT}\;\mathrm{agent}}{[P_{n}Z]\left(\sum_{m=1}^{N}k_{a}^{n,m}[P_{m}\cdot ]+k_{a}^{n,N}\mu_{0}^{N}(P\cdot )\right)}\;\;+\;\;\;\overset{\mathrm{intermediate}\;\mathrm{fragmentation}}{0.5k_{\beta }^{n}([P_{n}\overset{\cdot }{Z}P]+[{P\overset{\cdot }{Z}P}_{n}])} \end{array}$$

That for macroRAFT agent of length *n* > *N* is shown in Equation (15)
(15)$$\begin{array}{ll} \frac{d\mu_{x}^{N}(\mathrm{PZ})}{dt} & =\overset{\mathrm{reversible}\;\mathrm{photoinitiation}}{\left(-k_{\mathrm{dzp}}^{}\mu_{x}^{N}(\mathrm{PZ})+k_{\mathrm{tzp}}^{}\mu_{x}^{N}(P\cdot )[Z\cdot ]\right)}\;\;+\;\;\;\overset{\mathrm{transfer}\;\mathrm{to}\;\mathrm{disulfide}}{k_{\mathrm{trpZ}2}^{}\mu_{x}^{N}(P\cdot ){[Z}_{2}]}\\ & \overset{\;\mathrm{reaction}\;\mathrm{of}\;I\cdot \;\mathrm{with}\;\mathrm{macroRAFT}\;\mathrm{agent}}{-k_{-\beta ,I}^{}\mu_{x}^{N}(\mathrm{PZ})[I\cdot ]+k_{\beta ,I}^{}\mu_{x}^{N}(P\overset{\cdot }{Z}I)}-\overset{\mathrm{reaction}\;\mathrm{of}\;R\cdot \;\mathrm{with}\;\mathrm{macroRAFT}\;\mathrm{agent}}{k_{-\beta ,R}^{}\mu_{x}^{N}(\mathrm{PZ})[R\cdot ]+k_{\beta ,R}^{}\mu_{x}^{N}(P\overset{\cdot }{Z}I)}\\ & -\mu_{x}^{N}\overset{\mathrm{reaction}\;\mathrm{with}\;\mathrm{macroRAFT}\;\mathrm{agent}}{(\mathrm{PZ})\left(\sum_{m=1}^{N}k_{a}^{N,m}[P_{m}\cdot ]+k_{a}^{N,N}\mu_{x}^{N}\mu_{0}^{N}(P\cdot )\right)}\\ & +\overset{\mathrm{intermediate}\;\mathrm{fragmentation}}{0.5k_{\beta }^{}(\mu_{x}^{N}{(P}_{n}\overset{\cdot }{Z}P)+\mu_{x}^{N}{(P\overset{\cdot }{Z}P}_{n}))} \end{array}$$

#### 3.1.6. Dead Polymer Formed by Disproportionation ($P_{n}^{D}$) or Combination ($P_{n}^{C}$) of Propagating Radicals

The differential equation for dead polymer formed by disproportionation $P_{n}^{D}$ where the chain is of length ≤*N* is Equation (16).
(16)$$\frac{d[P_{n}^{D}]}{dt}=[P_{n}\cdot ]\left(\sum_{m=1}^{N}k_{\mathrm{td}}^{n,m}[P_{n}\cdot ]+k_{\mathrm{td}}^{n,N}\mu_{0}^{N}(P\cdot )\right)\;\;\;1\;<\;n\;\leq \;N$$

The differential equation for the partial moments $\mu_{x}^{N}(P_{}^{D})$ for chains length >*N* is Equation (17).
(17)$$\frac{d\mu_{x}^{N}(P_{}^{D})}{dt}=\sum_{n=1}^{N}k_{td}^{n,N}[P_{n}\cdot ]\mu_{x}^{N}(P\cdot )+k_{td}^{N,N}\mu_{0}^{N}(P\cdot )\mu_{x}^{N}(P\cdot )$$

In this work, we do not distinguish the polymer chains formed with saturated $P_{n}^{H}$ and unsaturated chain ends $P_{n}^{=}$, which will be formed in equal amounts ($P_{n}^{=}=P_{n}^{H}$). However, this may be important in circumstances where $P_{n}^{=}$ is reactive under the polymerization conditions, for example, as a comonomer in polymerization or as a macromonomer RAFT agent [22].

The differential equation for dead polymer formed by combination $P_{n}^{c}$ to form chains of length ≤*N* is Equation (18).
(18)$$\frac{d[P_{n}^{C}]}{dt}=0.5\sum_{m=1}^{n-1}k_{tc}^{n,m}[P_{m}\cdot ][P_{m-n}\cdot ]\;\;\;1\;<\;n\;\leq \;N$$

The differential equation for the partial moments describing dead polymer formed by combination $P_{n}^{c}$ to form chains of length >*N* is
(19)$$\frac{d\mu_{x}^{N}(P_{}^{C})}{dt}=0.5\sum_{n=N+1}^{\infty }n^{x}\sum_{m=1}^{n-1}k_{tc}^{m,n-m}[P_{m}\cdot ][P_{n-m}\cdot ]$$

This equation can be broken into three terms as follows. The first term describes combination where both reacting chains P∙ are of chain length *n*
≤
*N*
(20)$$\sum_{m=1}^{N}\left[P_{m}\cdot \right]\sum_{j=1}^{m}k_{tc}^{m,N-j+1}{\left(N-j+m+1\right)}^{x}[P_{N-j+1}\cdot ]$$

The second term describes combination of a chain P· of chain length *n*
≤
*N* with a chain P· with length *n* > *N*
(21)$$\begin{array}{ll} \sum_{n=1}^{N}n^{x}\sum_{m=N+1}^{n}k_{tc}^{m,n-m}[P_{m}\cdot ][P_{n-m}\cdot ] & =\sum_{m=1}^{N}\left[P_{m}\cdot \right]\sum_{j=N+1}^{\infty }k_{tc}^{m,N}{\left(j+m\right)}^{x}[P_{j}\cdot ]\\ & =\sum_{m=1}^{N}k_{tc}^{m,N}[P_{m}\cdot ]\sum_{j=N+1}^{\infty }\sum_{t=0}^{x}\left(\begin{array}{c} x\\ t \end{array}\right)m^{t}j^{x-t}[P_{j}\cdot ]\\ & =\sum_{t=0}^{x}\left(\begin{array}{c} x\\ t \end{array}\right)\sum_{m=1}^{N}k_{tc}^{m,N}m^{t}[P_{m}\cdot ]\sum_{j=N+1}^{\infty }n^{x-t}[P_{j}\cdot ]\\ & =\sum_{t=0}^{x}\left(\begin{array}{c} x\\ t \end{array}\right)\sum_{m=1}^{N}k_{tc}^{m,N}m^{t}[P_{m}\cdot ]\mu_{x-t}^{N}(P\cdot ) \end{array}$$

The derivation of the third term describing combination of two chains P· of chain length *n > N* can be carried out in similar fashion.
(22)$$\begin{array}{ll} \sum_{n=N+1}^{\infty }n^{x}\sum_{m=N+1}^{n}k_{tc}^{m,n-m}[P_{m}\cdot ][P_{n-m}\cdot ] & =k_{tc}^{N,N}\sum_{j=N+1}^{\infty }[P_{j}\cdot ]\sum_{n=N+1}^{\infty }{\left(n+j\right)}^{x}[P_{n}\cdot ]\\ & =k_{tc}^{N,N}\sum_{j=n+1}^{\infty }[P_{j}\cdot ]\sum_{n=N+1}^{\infty }\sum_{t=0}^{x}\left(\begin{array}{c} x\\ t \end{array}\right)j^{t}n^{x-t}[P_{n}\cdot ]\\ & =k_{tc}^{N,N}\sum_{t=0}^{x}\left(\begin{array}{c} x\\ t \end{array}\right)\sum_{j=n+1}^{\infty }j^{t}[P_{j}\cdot ]\sum_{n=N+1}^{\infty }n^{x-t}[P_{n}\cdot ]\\ & =k_{tc}^{N,N}\sum_{t=0}^{x}\left(\begin{array}{c} x\\ t \end{array}\right)\mu_{t}^{N}(P\cdot )\mu_{x-t}^{N}(P\cdot ) \end{array}$$

Thus, the complete expression for the partial moment is given by Equation (23)
(23)$$\begin{array}{ll} \frac{d\mu_{x}^{N}(P_{}^{C})}{dt} & =0.5\left(\sum_{m=1}^{N}\left[P_{m}\cdot \right]\sum_{j=1}^{m}k_{tc}^{m,N-j+1}{\left(N-j+m+1\right)}^{x}[P_{N-j+1}\cdot ]\right.\\ & +\sum_{t=0}^{x}\left(\begin{array}{c} x\\ t \end{array}\right)\sum_{m=1}^{N}k_{tc}^{m,N}m^{t}[P_{m}\cdot ]\mu_{x-t}^{N}(P\cdot )\\ & +\left.k_{tc}^{N,N}\sum_{t=0}^{x}\left(\begin{array}{c} x\\ t \end{array}\right)\mu_{t}^{N}(P\cdot )\mu_{x-t}^{N}(P\cdot )\right) \end{array}$$

#### 3.1.7. MacroRAFT Intermediates (P_n_ŻP_m_)

In the case of the differential equations for the intermediate formed by reaction of a propagating species with a macroRAFT agent (P*_n_*ŻP*_m_*), we consider the intermediate as comprising two separate distributions rather than one joint distribution. This means the intermediate is described by 2*N* differential equations rather than *N*^2^ equations_._ Thus, intermediates with at least one chain 1 *≤* n *≤ N* are described by Equations (25) and (26).
(24)$$\frac{d{[P}_{n}\overset{\cdot }{Z}P]}{dt}=k_{a}^{}[P_{n}\cdot ]\left(\sum_{m=1}^{N}\left[P_{m}Z\right]+\mu_{0}^{}(\mathrm{PZ})\right)-k_{\beta }^{}[P_{n}\overset{\cdot }{Z}P]$$
(25)$$\frac{d[P\overset{\cdot }{Z}P{}_{n}]}{dt}=k_{a}^{}[P_{n}Z]\left(\sum_{m=1}^{N}\left[P_{m}\cdot \right]+\mu_{0}^{}(P\cdot )\right)-k_{\beta }^{}[{P\overset{\cdot }{Z}P}_{n}]$$

The differential equations [Equations (26) and (27)] describe the moments of the molar mass distribution for intermediates formed by reaction of a propagating species with a macroRAFT agent with at least one chain n *> N*.
(26)$$\frac{d\mu_{x}^{N}(P_{n}\overset{\cdot }{Z}P)}{dt}=k_{a}^{}\mu_{x}^{N}(P\cdot )\left(\sum_{m=1}^{N}\left[P_{m}Z\right]+\mu_{0}^{N}(\mathrm{PZ})\right)-k_{\beta }^{}\mu_{x}^{N}(P_{n}\overset{\cdot }{Z}P)$$
(27)$$\frac{d\mu_{x}^{N}({P\overset{\cdot }{Z}P}_{n})}{dt}=k_{a}^{}\mu_{x}^{N}(\mathrm{PZ})\left(\sum_{m=1}^{N}\left[P_{m}\cdot \right]+\mu_{0}^{N}(P\cdot )\right)-k_{\beta }^{}\mu_{x}^{N}({P\overset{\cdot }{Z}P}_{n})$$

The species P*_n_*ŻP and PŻP*_n_* are formed and used in identical amounts. They have been retained as separate species only to allow for the potential introduction of chain length dependent rate parameters.

In assessing the importance of (irreversible) intermediate radical termination (Figure 5), a series of additional terms of the form shown in Equations (28)–(31) along with corresponding terms for the partial moments need to be included for each radical species. Note that this treatment provides information on the concentration, average arm length and arm length distributions for the 3-arm star that would be formed by combination P*_n_*ŻP*_m_* with a propagating species P_n_∙ but does not directly provide the molar mass distribution of that 3-arm star.
(28)$$\mathrm{For}\;\frac{d{[P}_{n}\overset{\cdot }{Z}P]}{dt},\;\mathrm{the}\;\mathrm{term}\;\mathrm{is}\;-k_{\mathrm{tpp}}^{}[P_{n}\overset{\cdot }{Z}P]\left(\sum_{m=1}^{N}\left[P_{n}\cdot \right]+\mu_{0}^{}(P\cdot )\right)$$
(29)$$\mathrm{for}\;\frac{d{[P\overset{\cdot }{Z}P}_{n}]}{dt}\;\mathrm{it}\;\mathrm{is}\;-k_{\mathrm{tpp}}^{}[{P\overset{\cdot }{Z}P}_{n}]\left(\sum_{m=1}^{N}\left[P_{n}\cdot \right]+\mu_{0}^{}(P\cdot )\right)$$
(30)$$\mathrm{for}\;\frac{d{[P}_{n}\cdot ]}{dt}\;\mathrm{it}\;\mathrm{is}\;-k_{\mathrm{tpp}}^{}[P_{n}]\left(\sum_{m=1}^{N}\left[P_{n}\overset{\cdot }{Z}P\right]+\mu_{0}^{}(P_{n}\overset{\cdot }{Z}P)\right)$$
and for $\frac{d{[P}_{n}\mathrm{ZPP}]}{dt}$, the term is
(31)$$k_{\mathrm{tpp}}^{}[P_{n}\cdot ]\left(\sum_{m=1}^{N}\left[P_{m}\overset{\cdot }{Z}P\right]+\mu_{0}^{}(P_{m}\overset{\cdot }{Z}P)\right)+k_{\mathrm{tpp}}^{}\mu_{0}^{}(P\cdot )([P_{n}\overset{\cdot }{Z}P]+[{P\overset{\cdot }{Z}P}_{n}])$$

#### 3.1.8. (Macro)RAFT derived radical (Z∙)

For RAFT polymerization with direct photoinitiation (photoiniferter process), we include the reversible dissociation of the initial (RZ) and macroRAFT agents (P*_n_*Z). The reactions involving initiator derived RAFT agent (IZ) will only be involved in the case where an additional initiator is used. It is also very important to include reversible photodissociation of the disulfide (Z_2_). The concentration of Z∙ is described by Equation (32). We have not included initiation by Z∙ in the simulation and have no experimental evidence for this process in the experiments discussed. It is known, however, that initiation by Z∙ occurs in the photoiniferter process [65] where polymerization may be initiated by a photodissociation of a dithiuram disulfide.
(32)$$\begin{array}{ll} \frac{d[Z\cdot ]}{dt} & =\overset{\mathrm{reversible}\;\mathrm{photodissociation}\;\mathrm{of}\;\mathrm{RZ}}{\left(k_{\mathrm{dzi}}^{}[\mathrm{RZ}]-k_{\mathrm{tzi}}^{}[R\cdot ][Z\cdot ]\right)}+\overset{\mathrm{transfer}\;\mathrm{to}\;\mathrm{disulfide}}{k_{\mathrm{triz}2}^{}[R\cdot {][Z}_{2}]}\\ & \overset{\mathrm{reversible}\;\mathrm{photodissociation}\;\mathrm{of}\;\mathrm{IZ}}{+\left(k_{\mathrm{dzi}}^{}[\mathrm{IZ}]-k_{\mathrm{tzi}}^{}[I\cdot ][Z\cdot ]\right)}+\overset{\mathrm{transfer}\;\mathrm{to}\;\mathrm{disulfide}}{k_{\mathrm{triz}2}^{}[I\cdot {][Z}_{2}]}\\ & +\sum_{n=1}^{N}\left[\overset{\mathrm{reversible}\;\mathrm{photodissociation}\;\mathrm{of}\;\mathrm{PZ}}{\left(k_{\mathrm{dzp}}^{}[P_{n}Z]-k_{\mathrm{tzp}}^{}[P_{n}\cdot ][Z\cdot ]\right)}+\overset{\mathrm{transfer}\;\mathrm{to}\;\mathrm{disulfide}}{k_{\mathrm{trpZ}2}^{}[P_{n}\cdot {][Z}_{2}]}\right]\\ & +\overset{\mathrm{reversible}\;\mathrm{photodissociation}\;\mathrm{of}\;\mathrm{PZ}}{\left(k_{\mathrm{dzp}}^{}\mu_{0}^{N}(\mathrm{PZ})-k_{\mathrm{tzp}}^{}\mu_{0}^{N}(P\cdot )[Z\cdot ]\right)}-\overset{\mathrm{transfer}\;\mathrm{to}\;\mathrm{disulfide}}{k_{\mathrm{trpZ}2}^{}\mu_{0}^{N}(P\cdot ){[Z}_{2}]}\\ & \overset{{\mathrm{reversible}\;\mathrm{photodissociation}\;\mathrm{of}\;Z}_{2}}{+\left(k_{\mathrm{dZ}2}^{}{[Z}_{2}]-k_{\mathrm{tZ}2}^{}{[Z\cdot ]}^{2}\right)} \end{array}$$

### 3.2. Kinetic Simulation

A series of experiments was conducted to explore the kinetics of thermally-initiated RAFT oligomerization [62] and photoRAFT oligomerization [61] of DMAm under various conditions. Numerical simulation was then used to estimate the rate coefficients associated with the RAFT equilibria for experiments conducted with [DMAm]:[RAFT]~2:1. 

In a previous paper [62] we reported the results of kinetic simulation was conducted using Predici. Both Predici modelling and the present method of partial moments yield equivalent results with the same kinetic parameters. In the earlier effort [62] we used a substantially lower (by ca two orders of magnitude) value for the DMAm propagation rate coefficient than we use in the present work (vide infra).

#### 3.2.1. RAFT Oligomerization of DMAm Thermally Initiated with a Dialkyldiazene

The reaction scheme is shown in Figure 6. In numerical simulation the thermally initiated oligomerization of DMAm the main task was to estimate the rate coefficient for the first propagation step and the kinetic parameters associated with the RAFT equilibrium. The values of rate parameters that are known from the literature or which could be reasonably estimated on the basis of literature data are discussed in Section 3.2.1.1, Section 3.2.1.2, Section 3.2.1.3 and Section 3.2.1.4. and summarized in Table 1. The outcome of numerical simulation with respect to modelling the RAFT oligomerization described in Section 2.1, and for which the experimentally determined evolution of various species with time is shown in Figure 1a, is presented in Figure 1b.

Reactions associated with the initialization process for conversion of the initial RAFT agent (RZ) to a macroRAFT agent (P_1_Z) are shown in Figure 6.

Several features of the process are worth noting.
Selective initialization is observed; i.e., there is no significant formation of dimer (P_2_∙) or higher oligomers until the initial RAFT agent (**1**) is largely consumed and the value of $k_{p}^{1}$ [DMAm] exceeds $k_{a,R}^{1}$ [1].The rate determining step in the consumption of the initial RAFT agent is rate of reinitiation by R·, *k*_iR_ is substantially lower than *k*_p_. The length of the initialization period is thus determined by the rate coefficients *k*_iR_ and the relative concentrations of monomer and RAFT agent.It is necessary to include intermediate radical termination in the simulation.

Selective initialization can be understood as follows. There is no significant concentration of the unimer radical P_1_∙ until the initial RAFT agent (RZ) is fully consumed. While some RZ remains, P_1_∙, when formed, is immediately and rapidly transformed to the unimer RAFT agent, P_1_Z, and R∙ by the RAFT with RZ. The transfer coefficient of RZ is sufficiently high such that on average less than one monomer unit is added per activation cycle. The unimer RAFT agent, P_1_Z, has a much lower transfer coefficient than RZ. Because R∙ is the better homolytic leaving group, any reaction of R∙ with P_1_Z is not productive and the intermediate formed rapidly reverts to R∙ and P_1_Z. Only when RZ is largely consumed, is there significant possibility for P_1_∙ being converted to P_2_∙ and higher oligomers by propagation.

##### 3.2.1.1. Propagation Rate Coefficients

There have been several studies on the propagation kinetics of DMAm [66,67,68] and other acrylamides. However, none of these studies specifically relate to very short chains. In 1978, Yamada et al. [67] used the rotating sector method to determine *k*_p_ and *k*_t_ for DMAm at 30 °C in bulk monomer as 2.72 × 10^4^ M^−1^s^−1^ and 3.54 × 10^9^ M^−1^s^−1^, respectively. More recently, the propagation kinetics for DMAm in aqueous solution were studied in detail by Schrooten et al. [68] using pulsed laser photolysis (PLP). They found a strong dependence of *k*_p_ on the weight fraction of monomer, and also found that *k*_p_ was substantially higher in aqueous solution than in bulk monomer. Their [68] expression for *k*_p_ is given in Equation (33).
(33)$$k_{p}\left(w_{\mathrm{DMAm}},\;T,\;p\right)=3.24\times 10^{7}\left(0.534+\left(1-0.534\right)e^{-9.78w_{\mathrm{DMAm}\;}}-0.410w_{\mathrm{DMAm}\;}\right)e^{-\frac{1.49\times 10^{4}-1.27p}{8.31441T}}{\mathrm{Lmol}}^{-1}s^{-1}$$
where *w*_DMAm_ is the weight fraction of DMAm, *p* is the pressure in bar, and *T* is the temperature in K. This suggests that the long chain value for *k*_p_ under our conditions (*w*_DMAm_ ~0.19, 60 °C, ~1 bar) should be *k*_p_(*n*) ~ 9.99 × 10^4^ L mol^−1^ s^−1^. The outcome is not dramatically affected by values of *k*_p_(>2).

The value of *k*_p_(1) then needed to be chosen to be sufficiently higher that *k*_p_(*n*) such that the ratio of 2-unit chains to higher oligomers matched experiment. Many authors have reported on chain length dependence of *k*_p_ for the first few propagation steps [69,70,71]. Values of *k*_p_(1) and *k*_p_(2) are generally found to be higher than *k*_p_(*n*). However, none of these studies relate to DMAm or other acrylamides. In the present simulations we use values of *k*_p_(≥1) that are consistent with the experimental *k*_p_(*n*) of Schrooten et al. [68] and a value of *k*_p_(1) = 7 × *k*_p_(2).The simulation is not strongly influenced by the values of *k*_p_(≥2).

##### 3.2.1.2. Dialkyldiazene Initiation Rate Coefficients

The value of *k*_d_ for VA044 in water (pH 7.5 buffer) is reported as 8.07 × 10^−5^ s^−1^ at 60 °C [log_10_(A/s^−1^) 12.64±0.08, *E*_a_ 106.7±0.5 kJ mol^−1^] [64]. The rate coefficient is known to be pH dependent. We determined *k*_d_ for VA044 in 0.045 M Na_2_CO_3_ in D_2_O at 60 °C directly from the observed rate of disappearance of the initiator from the ^1^H NMR spectra as 5.78 × 10^−5^ s^−1^. 

No relevant values of *k*_i_ for VA044-derived radicals have been reported. The value used is the same as *k*_iR_ for initiation by a tertiary cyanoalkyl radical (2-cyanopropan-2-yl radical) [64,69], which we expect is a reasonable approximation.

The value of *k*_iR_ is critical to determining the rate of utilization of the initial RAFT agent and was chosen to fit our experimental data for loss of RAFT agent. The value of *k*_iR_ for DMAm so estimated is similar to that reported for 2-cyanopropan-2-yl radicals adding to methyl acrylate [64,69].

##### 3.2.1.3. RAFT Rate Coefficients

The extent of propagation during the initialization step is determined by the values of *k*_add_ and *k*_p_(1). If *k*_p_(1)*:k*_add_ is too high for the concentrations of DMAm and RAFT agent used, then P_1_· will undergo multiple propagation steps rather than being trapped as the unimer by reaction with RAFT agent. To meet these conditions with *k*_p_(*n*) ~ 79000, the value of *k*_add_ was chosen as 2 × 10^8^ M^−1^s^−1^.

Values of *k*_-add_ and *k*_β_ are also important in determining the value of the transfer coefficient ($k_{\mathrm{tr}}=k_{\mathrm{add}}(k_{\beta }/k_{\beta }+k_{-\mathrm{add}})=k_{\mathrm{add}}\phi$). The occurrence of the back reaction will reduce the effective transfer coefficient. Similarly, *k*_-tr_ will reduce the effective transfer coefficient.

##### 3.2.1.4. Termination Rate Coefficients

Yamada et al. [67] reported a value of average termination rate coefficient, <*k*_t_>, for DMAm at 30 °C in bulk monomer obtained by the rotating sector method as 3.54 × 10^9^ M^−1^s^−1^. This value seems remarkably high relative to values of *k*_t_ in the literature for other polymerizations and with respect to the Smoluchowski model [72,73,74]. There are no reported termination coefficients relevant to oligomeric DMAm chains. We have chosen to use a value of k_t(i,j)_ suggested by the Smoluchowski model (for short chains) and the geometric mean model (for longer chains) – Equation (34).
(34)$$2\pi (D_{i}+D_{j})/\sigma N_{a}+kt_{0}{\left(i.j\right)}^{\left(-\beta /2\right)}$$
where *D_i_* is the diffusion coefficient for a chain of length *i* and is approximated as $D_{i}=D_{\mathrm{monomer}}/i^{\alpha }$ with *D*_monomer_ = 1.5 × 10^−9^ m^2^ s^−1^, *σ* is a capture radius (3 × 10^−8^ m), *N*_a_ is Avogadro’s number, and α and β are constants (*α* = 0.5, *β* = 0.65). 

The precise value of the termination rate coefficient does affect the rate of reaction, but otherwise has no dramatic effect on the results for the conditions explored.

##### 3.2.1.5. Intermediate Radical Termination Rate Coefficients

We see no reason that rate coefficients associated with intermediate radical termination (Figure 5) should not be diffusion-controlled and similar to those for other processes for radical-radical termination (Section 3.2.1.4). These rate coefficients would also be anticipated to show the same form of chain length dependence as those for other forms of termination. The significance of intermediate radical termination depends strongly on the lifetime of the intermediates. The process has greater significance when becomes of greater significance when rates of intermediate fragmentation are low and values of the reverse transfer constant (*C*_-tr_ = *k_-__β_/k*_iR_) are high. In order to successfully simulate our experimental results for RAFT oligomerization it was necessary to include intermediate radical termination, slow fragmentation in the simulation or both. 

We close value similar to those for other processes for radical-radical termination (Section 3.2.1.4), then chose values for the fragmentation rate coefficients that provided enabled fitting the observed concentration profile for the oligomeric RAFT agent shown Figure 1 (thermal initiation) and Figure 2 (photoRAFT initiation). Even with high rate coefficient for intermediated radical termination, the predicted concentration of products from intermediate radical termination was < 0.5% relative to the desired macro RAFT agent, well below the limits of detection in our NMR experiments. The major products predicted to arise from intermediate radical termination are P*_m_*Z(P*_n_*)P*_o_* and P*_m_*Z(P*_n_*)I or the corresponding disproportionation products.

It was also possible to achieve an acceptable to the experimental data with all of the rate coefficients for intermediate radical termination set to zero. However, in this case the concentration of intermediate radicals (in particular, P_1_ŻP_1_) is predicted to reach the quite unrealistic value of 0.01 M.

Other recent work attests to the ubiquitous nature of retardation in RAFT polymerization and the inability to discriminate models using kinetic data [21,59]. 

#### 3.2.2. Thermally Initiated RAFT Polymerization of *N*,*N*-Dimethylacrylamide (DMAm)

With the various rate parameters more or less established we decided to explore what the selected rate parameters meant for a DMAm polymerization under similar conditions, but with a higher [monomer]:[RAFT] agent ratio. The predicted evolution of molar mass and dispersity of all chains and of living (macroRAFT) and dead chains with time and monomer conversion for a conventionally initiated polymerization with [monomer]:[RAFT agent]:[VA-044] = 5:0.1:0.01 are shown in Figure 4. The values for total chains overlap those for the macroRAFT chains. The predicted variation of species vs. time is shown in Figure 5 and the SEC molar mass distributions for the macro RAFT agent and dead polymer are shown in Figure 6. A small dependence of the output on changing the value of *N*, is seen because of the use of chain length dependent termination rate coefficients *k*_t_. The following points should be noted. 

The dispersity of macroRAFT agent is ca 1.05 at ca 50 min, which corresponds to ~100% monomer conversion. For longer times the dispersity slowly increases as propagating radicals are still being generated by RAFT and still undergoing termination.An inhibition period is observed that corresponds to the time taken to convert the initial RAFT agent to macroRAFT agent. A tertiary cyanoalkyl RAFT agent is not ideal for this polymerization as the rate of addition of the cyanoalkyl radicals to monomer is rate determining. Nonetheless, there is a near linear correspondence between molar mass and conversion as anticipated for a well-behaved RDRP.Use of an initial RAFT agent that resembled the macroRAFT agent (*n* = 1) would give complete conversion in ca 25 min.The amount of termination (fraction of dead chains formed) is very small such that the distributions for total polymer and macroRAFT agent overlap. This can be appreciated by considering the Y axis scales of Figure 6a,b (in ratio 1:0.025). The dead chains lie at the origin in Figure 5.Termination between propagating species is assumed to occur by combination. The molar mass distribution of dead polymer for long times approaches that for macroRAFT agent as it is mostly formed by primary radical termination.

#### 3.2.3. Direct photoRAFT Oligomerization of DMAm

In direct photoRAFT process radicals are produced directly by photodissociation of the RAFT agent. There is no separate initiator. The additional reactions used to describe photoRAFT are shown as Reactions (31–41) in Figure 2. The initialization process for conversion of the initial RAFT agent to a macroRAFT agent is shown in Figure 7.

As an initial guess, the rate coefficients for photodissociation of the initial RAFT agent (ZR), macroRAFT agent (ZP*_n_*) and the disulfide Z_2_ were set at the same value and give the observed rate of disappearance of the initial RAFT agent. The value for the macroRAFT agent ZP_n_ was then adjusted to give the observed rate of unimer disappearance. The value for the unimer macroRAFT agent needed to be ~3 orders of magnitude lower than that for the initial RAFT agent. 

The rate coefficients for the RAFT equilibrium were initially set to be the same as used to model RAFT oligomerization with thermal initiation. However, we found it necessary to have a ~ 5-fold increased rate coefficient for addition of R∙ to RAFT agent and for intermediate radical termination involving propagating radicals P*_n_*∙ to achieve the result shown in Figure 2b.

There is little information on rates coefficients for reactions of the thiocarbonylthio radicals, Z∙. Kuchanov [75] has indicated that rate coefficients for deactivation by reaction with diththiocarbamyl radicals R_2_NCS_2_∙ (*k*_tiz_, *k*_trz_, *k*_tpz_) are diffusion-controlled so we might anticipate that those for other Z∙ should also be diffusion-controlled and therefore similar to *k*_prt_.

It is notable that photoRAFT oligomerization (and polymerization), unlike RAFT with an exogenous initiator), is subject to the persistent radical effect.

The radical Z∙ has very low, though not negligible, reactivity towards monomer, and may be subject to side reactions such as dithiodecarboxylation. It is expected to undergo self-reaction to form the disulfide [76] possibly also with a diffusion controlled rate coefficient [75]. This reaction mitigates against the influence of the persistent radical effect [77,78].

We used simulation to explore the use of faster photodissociation sufficient to give a rate of disappearance similar to that seen in the thermally initiated experiment. The model could in principle be expanded to cover PET-RAFT oligomerization of DMAm, which provides for better selectivity and higher rates of reaction than direct photoinitiation [79]. However, the detailed mechanism of initiation is not yet established and there is little guide as to the values of the rate parameters associated with the process [80,81,82]. Accordingly, this is left for future study.

#### 3.2.4. Direct photoRAFT Polymerization of DMAm

Direct photoRAFT polymerization DMAm with [monomer]:[RAFT] 5:0.1 was then explored. The reaction conditions emulated in the simulation were similar to those to the thermally initiated experiment except for the absence of an initiator and the use of (nominally blue light) irradiation. The same rate parameters for propagation, termination and the RAFT process as used in other simulations were used. The rate of photodissociation of the initial RAFT agent was chosen so as to give a similar lifetime as seen in the thermally initiated experiment. Other photo dissociation rates were increased proportionately. The outcome is presented in Figure 4, Figure 5, Figure 6, Figure 7, Figure 8 and Figure 9. Points arising from the simulation.
The amount of dead polymer formed by bimolecular termination is significantly smaller than that formed in the above experiment with conventional initiation. The dominant process producing dead polymer is intermediate radical termination. This is not included in the plots below. The amount of intermediate radical termination equates to an amount of disulfide (ZCS_2_)_2_ formed.The dispersity of the macro RAFT agent and total polymer (*Đ* ~ 1.03, for > 50% conversion Figure 7) are significantly lower than that formed in the above experiment with conventional initiation.

## 4. Conclusions

We have described a method of partial moments for kinetic simulation of the time/conversion evolution of the molar mass distributions in radical polymerization. The method provides a complete description of the molar mass distribution for components with a degree of polymerization < 200, while components with higher degrees of polymerization are characterized only in terms of the partial moments of the distribution.

We have applied the method to oligomers and polymers formed by RAFT polymerization. In particular, we have used the method to model and estimate rate coefficients in thermally- and photochemically-initiated RAFT oligomerization of DMAm. We stress that the rate parameters estimated are a set that provide a reasonable fit to the experimental data and are consistent with rate parameters that have been so far determined. The parameters may not be unique and actual rate parameters may differ. The rate parameters have then been used to predict the time/conversion evolution of molar mass distributions formed in conventionally initiated and direct photoRAFT initiated RAFT polymerization of DMAm.

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
