# Peer review of "Evolution of Molar Mass Distributions Using a Method of Partial Moments: Initiation of RAFT Polymerization"

_polymers, 2022, doi:10.3390/polym14225013_

Round 1
Reviewer 1 Report
polymers-1962968 – Evolution of Molar Mass Distributions using a Method of Partial Moments. Initiation of RAFT Polymerization.
1- The introduction could be improved by referring to the following refs including
https://doi.org/10.3390/polym10020224
https://doi.org/10.3390/nano9060828
https://doi.org/10.3390/polym14071448
https://doi.org/10.1002/macp.202000311
which can offer help to compose a better writing for the introduction and retrieve useful information regarding this work as well.
2- Analyses of data/results related to (section 2.2. Photochemically initiated RAFT...) hould be further elevated
3- Focus of aims should be further elevated in this work.
I recommend this work for publication after major revision.
Author Response
We have expanded the introduction to RDRP and RAFT polymerization to provide further background to the method and have included citation of the review mentioned by the referee.
Note that the first paper mentioned by the referee is already cited in our paper in the paragraph mentioning Predici simulation.
The other two papers mentioned by the referee have not been cited as they do not seem to be specifically relevant.
We have expanded (elevated) the section on photoRAFT to provide more detail and examples.
The focus of the work is to introduce a numerical simulation method that is specifically relevant to studying the intimate details of initiation of polymerization. We have attempted to make this clear in the abstract and introduction
Reviewer 2 Report
Dr.Moad et.al reported an interesting method of partial moments devised for accurate simulation of the time/conversion evolution of polymer composition and molar mass. This method is further applied to compare the behavior of thermally initiated and photo-initiated RAFT polymerization. The paper is well written and organized, and the work is well performed. Such an excellent paper should be very suitable for publication. I do not have many comments considering the high quality of this work. Just have some very minor suggestions.
1. Figure 5, x axis, there is an extra “”” before “times”. Besides, the title of y axis could be added.
2. The difference in mechanism between photo-raft (photochemical or photolysis) and thermal RAFT could be briefly introduced with one or two sentences, in order to help readers to better understand. Some recent examples/progress of photo-RAFT polymerization could be cited (https://doi.org/10.1021/acsmacrolett.0c00232, https://doi.org/10.1002/anie.202016523).
Author Response
Figure 5 has been fixed.
The section on photoRAFT initiation has been expanded to explain the difference from thermal initiation. Note that in this paper we only consider direct photo-initiation.
Round 2
Reviewer 1 Report
After this revision it in the manuscript ID "polymers-1962968" this paper is accept in present form and can be a very good contribution to polymers.